# The Impact of the COVID-19 Pandemic on Pediatric Microbial Resistance Patterns and Abandonment Rates in Western Romania—An Interdisciplinary Study

**DOI:** 10.3390/antibiotics14040411

**Published:** 2025-04-16

**Authors:** Dan Dumitru Vulcanescu, Iulia Cristina Bagiu, Tiberiu Liviu Dragomir, Virgiliu Bogdan Sorop, Mircea Diaconu, Octavia Harich, Sonia Aniela Tanasescu, Florin Szasz, Luiza Vlaicu, Cosmin Goian, Florin George Horhat

**Affiliations:** 1Doctoral School, “Victor Babes” University of Medicine and Pharmacy, Eftimie Murgu Square 2, 300041 Timisoara, Romania; dan.vulcanescu@umft.ro; 2Department of Microbiology, “Victor Babes” University of Medicine and Pharmacy, Eftimie Murgu Square 2, 300041 Timisoara, Romania; bagiu.iulia@umft.ro (I.C.B.); horhat.florin@umft.ro (F.G.H.); 3Multidisciplinary Research Center on Antimicrobial Resistance (MULTI-REZ), Microbiology Department, “Victor Babes” University of Medicine and Pharmacy, Eftimie Murgu Square 2, 300041 Timisoara, Romania; 4Clinical Laboratory, “Louis Turcanu” Emergency Hospital for Children, Timisoara, Iosif Nemoianu Street 2, 300011 Timisoara, Romania; 5Medical Semiology II Discipline, Internal Medicine Department, “Victor Babes” University of Medicine and Pharmacy, Eftimie Murgu Square No. 2, 300041 Timisoara, Romania; dragomir.tiberiu@umft.ro; 6Department of Obstetrics and Gynecology, “Victor Babes” University of Medicine and Pharmacy, 300041 Timisoara, Romania; bogdan.sorop@umft.ro (V.B.S.); diaconu.mircea@umft.ro (M.D.); 7Department of Functional Sciences, “Victor Babes” University of Medicine and Pharmacy Timisoara, 300041 Timisoara, Romania; harich.octavia@umft.ro; 8Department of Pediatrics, “Victor Babes” University of Medicine and Pharmacy, Eftimie Murgu Square No. 2, 300041 Timisoara, Romania; tanasescu.sonia@umft.ro; 9Department of Obstetrics and Gynaecology, Faculty of Medicine and Pharmacy, University of Oradea, 410073 Oradea, Romania; 10Department of Social Assistance, Faculty of Sociology and Psychology, Western University, Vasile Parvan Boulevard 4, 300223 Timisoara, Romania; florina.vlaicu@e-uvt.ro (L.V.); cosmin.goian@e-uvt.ro (C.G.)

**Keywords:** COVID-19, pandemic, microbial, trends, resistance, phenotype, Romania, abandonment

## Abstract

**Background:** The COVID-19 pandemic in Romania exacerbated pediatric antimicrobial resistance (AMR). Overuse of broad-spectrum antibiotics may be related to increased multidrug-resistant bacteria. The main aim of this study was to assess pediatric AMR trends and phenotypes, while a secondary objective was to investigate the potential links with hospital abandonment. **Methods:** This retrospective study from the Children’s Emergency Hospital “Louis Țurcanu”, Timișoara, focused on AMR patterns in 2019 pre-pandemic, 2021 pandemic, and 2023 post-pandemic. The following phenotypes were assessed: MRSA, MRCoNS, VRE, ESBL, CRO, MDR, XDR, and PDR. **Results:** There were 3530 total patients and 6885 total samples. There were 69.92% of the total samples resistant to at least one antimicrobial class, (72.69% in 2019, 67.05% in 2021, 69.16% in 2023). Specifically, resistance towards penicillins remained high across the entire period (57.45–60.93%), while the following classes presented elevated resistance in the pandemic: cephalosporins (42.91%), combination therapies (40.95%), reserve antibiotics (38.89%), and cyclines (13.83%). As for resistance phenotypes, MRSA and MRCoNS peaked during the pandemic (36.08% and 81.43%, respectively) while VRE remained relatively constant. Overall ESBL declined in 2023 to 14.45%, while overall CRO peaked during the pandemic (8.81%). Overall MDR fell during the pandemic (64.47%), while overall XDR peaked in 2019 (9.87%). No PDR cases were observed. Pediatric abandonment was an increasing concern, with regional cases rising from 5.42% (2019) to 9.83% (2023). Compared to the general population, increased antimicrobial resistance in abandoned patients was observed for fluoroquinolones (50.00%), Aminogycolsides (60.00%), reserve antibiotics (70.00%), cephalosporins (60.00%), and urinary antibiotics (60.00%). Resistance to cephalosporins (OR = 5.17, *p* = 0.0304) and reserve antibiotics (OR = 5.64, *p* = 0.0049) were key predictors of abandonment risk. **Conclusions:** The COVID-19 pandemic influenced resistance trends, with notable peaks in MRSA, MRCoNS, and CRO. Post-pandemic patterns suggest continued escalation of resistance. The association between resistant infections and pediatric abandonment highlights the need for robust antimicrobial stewardship and social intervention policies.

## 1. Introduction

The emergence of SARS-CoV-2 in late 2019 marked the onset of a global health crisis that rapidly evolved into a pandemic, officially declared by the World Health Organization (WHO) in March 2020 [1]. Initially, COVID-19 was believed to primarily affect adults and seniors, with children experiencing milder symptoms and lower hospitalization rates. However, as new variants emerged, pediatric transmission increased, highlighting the need for further investigation into the broader implications of the pandemic on child health and hospital care [2].

Beyond its direct impact on morbidity and mortality, COVID-19 reshaped medical, social, and economic landscapes. Romania, like many Eastern European nations, faced significant challenges during the COVID-19 pandemic due to its already strained healthcare infrastructure. The country experienced multiple waves of infection, with fluctuating case numbers, hospitalizations, and mortality rates, placing immense pressure on its limited intensive care unit (ICU) capacity. In pediatric care, pandemic disruptions resulted in reduced access to preventive medicine, vaccination programs, and routine check-ups, raising concerns about long-term health implications for children. The effects were particularly severe for socially vulnerable groups, including children from low-income families, institutionalized youth, and those with limited access to healthcare and education. The pandemic exacerbated malnutrition, delayed treatments for chronic illnesses, and heightened the risk of child abandonment and institutionalization. Additionally, the economic consequences of the crisis further deepened existing social vulnerabilities, intensifying the challenges faced by the most at-risk populations. Children represented a distinct patient group during the COVID-19 pandemic. While severe cases of COVID-19 were relatively rare among children compared to adults, pediatric patients remained vulnerable to indirect health consequences, including delayed medical attention, psychological distress, and secondary infection [3,4,5,6,7,8,9,10].

Although children accounted for a smaller proportion of confirmed cases in Romania, reports from the National Institute of Public Health and ECDC indicate that pediatric infections comprised approximately 10–12% of all COVID-19 cases during peak pandemic periods. Between 2020 and 2022, more than 80,000 pediatric cases were confirmed, with over 1200 hospitalizations and approximately 30 fatalities among individuals under 18, largely associated with pre-existing conditions [11].

While substantial research has focused on the epidemiology, clinical management, and vaccination strategies for COVID-19, certain critical aspects remain underexplored in Romania, particularly in the pediatric population. One key area is the impact of the pandemic on pediatric bacterial infections, antimicrobial resistance (AMR) patterns, and child abandonment rates in hospital settings. The widespread use of antibiotics to manage suspected bacterial co-infections in COVID-19 cases raised concerns about pediatric AMR, an issue exacerbated by the overuse and misuse of antimicrobial agents during the pandemic. [8,9,12,13,14,15]. Understanding shifts in pediatric AMR patterns is crucial for guiding future antibiotic stewardship programs and mitigating the long-term consequences of pandemic-era prescribing practices.

Antibiotic prescribing patterns revealed widespread reliance on broad-spectrum antibiotics, particularly penicillins, cephalosporins, and carbapenems, which may have contributed to accelerated AMR trends. A systematic review of 87 Romanian studies conducted during the pandemic found that up to 72.82% of bacterial isolates, particularly *Klebsiella* spp., *Pseudomonas* spp., and *Acinetobacter* spp., exhibited MDR, XDR, or pan-drug resistance (PDR) phenotypes. The increased prevalence of MRSA, vancomycin-resistant Enterococci (VRE), and ESBL-producing Gram-negative bacteria further complicates treatment strategies and underscores the urgent need for antimicrobial stewardship programs in pediatric care settings [16].

Child abandonment, a persistent social issue in Romania, was exacerbated by the COVID-19 pandemic due to worsening economic conditions, parental job losses, and reduced access to social support services. Romania already had one of the highest child abandonment rates in Europe, an issue which was worsened by the pandemic. Pediatric hospital abandonment, a complex issue influenced by medical, social, and economic factors, was similarly affected [17]. Additionally, disruptions in adoption processes and foster care placements due to lockdowns and bureaucratic slowdowns resulted in prolonged stays in state institutions, further impacting children’s development and well-being. Conversely, restricted hospital admissions during the pandemic may have led to an underreporting of abandonment cases. Some children who might have otherwise been left in hospitals were not admitted due to limited access to healthcare services, masking the true extent of the issue [18,19]. Understanding these shifts is crucial for assessing the long-term impact of the pandemic on pediatric abandonment and developing targeted social interventions to address these vulnerabilities.

In the end, this study addresses two critically underexplored and potentially interlinked challenges in Romanian pediatric healthcare: the evolution of AMR patterns and pediatric abandonment during the COVID-19 pandemic. While AMR is a growing global health threat, regional disparities and pandemic-related disruptions have amplified its effects in vulnerable settings like Western Romania. At the same time, pediatric abandonment may be exacerbated by increased healthcare burdens and social stressors, including infection severity and length of hospital stays. By combining microbiological and social data across three pandemic timepoints, this study offers a unique interdisciplinary perspective. Its findings not only reveal shifting resistance dynamics but also identify resistant infections as potential predictors of abandonment, suggesting an urgent need for integrated antimicrobial and social care policies.

As such, the objectives of this paper are as follows:To analyze the trends in pediatric AMR patterns in Romanian hospitals before, during, and after the pandemic.To assess the impact of the pandemic on hospital abandonment rates among pediatric patients, particularly in Western Romania.To examine potential correlations between bacterial infections, AMR trends, and hospital abandonment cases, assessing whether infection severity influenced abandonment risks.To compare national, county, and hospital-level trends, identifying regional disparities and healthcare inequalities.To raise awareness on the subject of pediatric abandonment healthcare units.To propose evidence-based strategies for mitigating AMR and improving child welfare policies in hospital settings.

## 2. Results

### 2.1. Overview, Demographics, and Infection Pofiles

Data pertaining to patient demographics such as sex, location of origin, and age can be observed in Appendix A. The pandemic year recorded the lowest number of patients and samples compared to both pre-pandemic (2019 vs. 2021 patient ratio was 1.85, *p* < 0.0001; 2019 vs. 2021 sample ratio was 1.49, *p* < 0.0001) and post-pandemic (2023 vs. 2021 patient ratio was 1.95, *p* < 0.0001; 2023 vs. 2021 sample ratio was 1.63, *p* < 0.0001) periods. Additionally, the post-pandemic year had a higher sample count than the pre-pandemic year (2019 vs. 2023 ratio was 0.91, *p* = 0.0006). Further, demographic data are presented in regards to the number of patients. The proportion of male patients (51.25%) was slightly higher, but this difference was not statistically significant (*p* = 0.1431). Additionally, when assessing sex distribution across the years using a Chi-square test, no significant variation was found (*p* = 0.2489). In terms of the location of origin, a higher proportion of patients came from urban areas (53.71%), with a statistically significant result (*p* < 0.0001). The Chi-square test for location of origin across the years showed a significant difference (*p* = 0.0028), with urban-origin patients being more prevalent in 2019 and 2021, while in 2023, the distribution between urban and rural patients was nearly equal. Regarding age groups, the following categories were identified: infant (<2 years), preschool (2–5 years), school (6–12 years), and adolescent (12–18 years), with infants being the most populous category, as confirmed by the Chi-square test (*p* = 0.0334). However, when considering age as a continuous variable, no significant differences were observed using the Kruskal–Wallis test, with the median age being 3 years for all three studied years.

Ward representation can be observed in Appendix A. Key observations included the following: The Outpatient Department had the highest patient volume across all years (23.34% of total patients), though numbers dropped during the pandemic (2021: 20.52%) before rising again post-pandemic (2023: 21.76%). The ICU (13.17%) and Surgery (11.25%) had significant patient loads, with admissions peaking during the pandemic (18.56% in 2021 for the ICU and 15.56% for Surgery). While Dialysis also peaked during the pandemic (1.31%), it declined after (0.34%), and NICU, Preterm, and Neonatology cases were relatively stable.

The median LoS was 3 days in 2019, 5 days in 2021, and 2 days in 2023 (*p* < 0.0001), with post-hoc Dunn’s tests confirming significant differences between all pairwise comparisons. For ICU/NICU patients, the median LoS was 8 days in 2019, 10 days in 2021, and 9 days in 2023 (*p* = 0.0171), with a statistically significant difference between 2019 and 2021 (*p* = 0.0044), while comparisons between 2019 vs. 2023 (*p* = 0.2447) and 2023 vs. 2021 (*p* = 0.1333) were not significant. These findings suggest that hospital stays were longest during the pandemic year (2021), while post-pandemic hospitalizations (2023) had the shortest durations.

Further, data are presented in regards to the number of samples, for which the distribution can be seen in Appendix A. Key observations included the following: The most frequent samples were urine (28.85%), nasal secretions (13.09%), and wound secretions (11.20%). While urine maintained a relatively consistent collection rate across the studied years, during the pandemic, there was a significant drop in nasal secretion (15.18% to 10.41%), while wound secretions increased (8.86% to 15.85%). Stool and skin samples remained consistently low across all years, while pharyngeal exudate rates dropped heavily in the pandemic (8.10% to 1.14%), which resurged to 12.34% in 2023.

Data on pathogen identification are presented as an overview and are observable in Appendix A. Pathogen distribution trends were analyzed in a separate study and are not within the scope of the present analysis. Key observations included the following: The most frequently identified microorganism across all samples was *Escherichia coli* (16.99%), with a relatively stable prevalence across the years, followed by *Staphylococcus aureus* (13.96%), which observed a decline in 2023 (15.25% to 12.67%); *Pseudomonas aeruginosa* (8.83%), which peaked during the pandemic (9.02% to 11.18%) and dropped afterwards in 2023 (7.22%); *Klebsiella pneumoniae* (8.63%), also relatively stable; and group A *Streptococcus* (8.51%, which nearly disappeared during the pandemic (7.69% to 0.84%), with a dramatic resurgence in 2023 (13.96%). Other important increases during the pandemic were in regards to *Candida albicans* (4.91% to 7.72%) and *Stenotrophomonas maltophilia* (1.77% to 5.08%), while CoNS and other *Enterococcus* spp. increased post-pandemic (4.19% to 6.96% and 1.02% to 3.04%, respectively).

### 2.2. Tested Antimicrobial Trends

We have noticed a significant shift in antimicrobial agents used for susceptibility testing, both in terms of class and individual antibiotics when pooled together. The frequency rates for each individual agent and classes can be found in Appendix A. From these tables, it is noticeable that the most prevalent antibiotics used for susceptibility testing were fluoroquinolones (93.07%), aminoglycosides (90.04%) and penicillins (88.40%).

For each class and individual antibiotic, a crosstab was created between the years and the susceptibility testing results. The crosstabs were then tested using the Chi^2^ test, and the results can be observed in Table 1. The following antibiotics could not be tested, as they had entries for only one year: Amoxicillin, Ceftizoxime, Cefuroxime, Neomycin, Clarithromycin, Clotrimazole, and Voriconazole. The following tests were performed only between two periods, with the third one missing: Ceftazidime/Avibactam, Ofloxacin, Minocycline, and Caspofungin. The majority of the Chi^2^ test results were statistically significant both in terms of class and individual antimicrobial properties, while the following individual drugs showed no statistically significant results: Netilmicin (*p* = 0.3366), Minocycline (*p* = 0.6935), vancomycin (*p* = 0.6429), Nalidixic Acid (*p* = 0.8504), Amphotericin B (*p* = 0.0624), Fluconazole (*p* = 0.0908), and Caspofungin (*p* = 0.0882).

As some antimicrobials were either not tested in certain years or had fewer tests than anticipated, further analysis was performed by classes. Regarding samples that exhibited resistance, 4814 (69.92%) of the total were resistant to at least one class overall, with 1805 (72.69%) in 2019, 1121 (67.05%) in 2021, and 1888 (69.16%) in 2023. Data on frequency regarding class resistance can be observed in Appendix A. The incidence rate ratio comparison between 2019 and 2021 samples was statistically significant (*p* = 0.0330), the incidence in 2019 being 1.08 times higher than that of 2021, while the results between 2019 vs. 2023 and 2021 vs. 2023 were not statistically relevant (*p* = 0.1297, 0.4108, respectively). The comparison of rates is observable in Table 2. The following statistically significant changes can be observed: an increase in combination therapy and reserve antibiotics; a decrease in glycopeptides, urinary antibiotics, and all antifungals between 2019 and 2021; a decrease in cephalosporins, carbapenems, aminoglycosides, lincosamides, and glycopeptides; an increase in fluoroquinolones, urinary antibiotics, antibiotics classified as “Other”, polyenes, and other antifungals between 2019 and 2023; a decrease in cephalosporins, combination therapy, carbapenems, aminoglycosides, macrolides, lincosamides, cyclines, and reserve antibiotics; and an increase in fluoroquinolones, urinary antibiotics, other antibiotics, and all antifungals.

### 2.3. Resistance Phenotype Analysis

The following resistance phenotypes were studied according to well-defined criteria: MRSA, MRCoNS, VRE, ESBL, CRO, MDR, and XDR [20,21,22,23]. The following totals were observed: MRSA—185 (19.25%), MRCoNS—182 (48.02%), VRE—13 (3.38%), ESBL-—1323 (19.22%), CRO—350 (5.08%), MDR—4690 (68.12%), and XDR—545 (7.92%). Overall, the most important ESBL producers were *Escherichia coli* (32.96%), *Klebsiella* pneumonias (28.72%), and *Pseudomonas aeruginosa* (13.53%), while for carbapenemase producers, they were *Pseudomonas aeruginosa* (49.14%), *Klebsiella pneumoniae* (22.57%), and *Chryseobacterium* spp. (6.00%). Among MDR strains, the most frequently isolated bacteria included *Escherichia coli* (19.91%), *Staphylococcus aureus* (13.82%), and *Klebsiella pneumoniae* (13.82%), while among XDR strains, the following top strains were observed *Klebsiella pneumoniae* (37.61%), *Pseudomonas aeruginosa* (18.72%), and CoNS (15.23%). Detailed incidence data can be observed in Appendix A. The incidence of these phenotypes for each bacterium/fungus was reported to the total strains of the respective microorganism, and then, the incidence rates were compared using the IRR; the results of the IRR testing can be observed henceforth.

Regarding trends for MRSA, it is observable that there was an increase in 2021, compared to both pre- and post-pandemic levels, while a similar trend was observable for MRCoNS, for which the difference between 2019 and 2023 was also statistically significant, resulting in a 1.57 times higher incidence in 2019. VRE IRR failed to reach statistical significance. The specific IRR values are observable in Table 3.

Regarding the ESBL phenotype, overall, there was a decrease in the trend between 2019 and both 2021 and 2023. Specific trends were as follows: a decrease in *Escherichia coli* between 2019 and both 2021 and 2023, a decrease in *Pseudomonas aeruginosa* between 2021 and 2023, and a decrease between both 2019 and 2021 as compared to 2023 in *Serratia marcescens* and *Enterobacter* spp. The following bacteria did not present ESBL phenotype across two of the studied timeframes: *Chryseobacterium* spp., *Sphingomonas paucimobilis*, other *Serratia* spp., bacteria designated as “Other”, and other *Acinetobacter* spp. Exact values for the IRR are presented in Table 4.

Overall carbapenemase producer rates showed a peak during the pandemic, followed by 2019. This trend was also followed in *Pseudomonas aeruginosa* and *Enterobacter* spp. The trend for *Klebsiella pneumoniae* showed an increase in 2023 as compared to both pre-pandemic and pandemic levels. The following comparisons could not be established fully due to missing percentages across two years: *Chryseobacterium* spp., *Sphingomonas paucimobilis*, other *Pseudomonas* spp., other *Acinetobacter* spp., and *Serratia marcescens*. The following comparisons could not be established due to extremely low counts: *Morganella* spp. and other *Proteus* spp. These data are observable in Table 5.

Overall MDR strains featured a descending peak during the pandemic, with the incidence returning to pre-pandemic levels afterwards. More specifically, for *Escherichia coli*, an increasing trend was observed between 2019 and 2023; for *Staphylococcus aureus*, an increase in 2023 as compared to both 2019 and 2021; for *Pseudomonas aeruginosa*, a dip during the pandemic, with the highest frequency before the pandemic; for *Streptococcus pneumoniae*, a descending trendline between 2023 and both 2019 and 2021; for CoNS, a steady decrease between 2019 and both 2021 and 2023, for group A *Streptococcus* the peak in 2019 as compared to 2023; for *Enterobacter* spp., a peak during the pandemic, followed by a minimum in the post-pandemic period; a continued decrease for *Acinetobacter baumannii*, with 2023 being the minimum, as compared to both 2019 and 2021; a maximum incidence for *Stenotrophomonas maltophilia* and microbes labeled as “Other” in 2019 as compared to both 2021 and 2023; and a minimum in 2023 for Salmonella as compared to both 2019 and 2021. The following comparisons could not be established fully due to missing percentages across two years: *Chryseobacterium* spp., *Sphingomonas paucimobilis*, other *Serratia* spp., and other *Candida* spp. The following comparisons could not be established due to extremely low counts: *Candida parapsilosis*, *Candida tropicalis*, and *Haemophilus influenzae*. These data are observable in Table 6.

Data on XDR bacteria comparison are observable in Table 7. Overall rates observed a maximum in the pre-pandemic period as compared to both other timeframes. More specifically, however, *Klebsiella pneumoniae* observed a minimum during the pandemic as compared to both pre- and post-pandemic rates, while *Serratia marcescens* peaked in the pre-pandemic period only when compared to 2021. The following comparisons could not be established fully due to missing percentages across two years: *Chryseobacterium* spp. The following comparisons could not be established due to extremely low counts: *Acinetobacter baumannii*, *Morganella* spp., other *Pseudomonas* spp., other *Acinetobacter* spp., and microbes labeled as “Other”.

### 2.4. Interdisciplinary Research—Pediatric Abandoment in the Hospital

During 2019, there were a total of 683 reported cases of pediatric abandonment in healthcare units in the whole country, 388 in 2021 and 295 in 2023. In the Western region, there were 37 (5.42%) such cases in 2019, 29 (7.47%) in 2021, and 29 (9.83%) in 2023. On a county level, Timis observed 7 (1.02%) cases in 2019, 7 (1.80%) in 2021, and 10 (3.39%) in 2023. There were six (0.88%) abandoned pediatric patients during 2019, two (0.52%) during 2021, and five (1.69%) during 2023 at the “Louis Turcanu” Children’s Emergency Hospital, the main hospital for children in the Timis county and representative for the Timisoara city. Of these patients, there were four (0.59%) in 2019, zero (0.00%) in 2021, and three (1.02%) in 2023 who were diagnosed and confirmed with a microbial infection. The results of the rates of the abandonment analysis can be observed in Table 8 and show an increasing trend at the regional and county-wide levels but not for the hospital level. From the identified patients in the hospital, there were 10 samples in 2019 and 5 in 2023.

Demographic data for these patients can be seen in Appendix A, while a pathogen comparison can be found in Appendix A. There was a statistically significant difference in regards to 2023 *Stenotrophomonas maltophilia* incidence in abandoned patients when compared to non-abandoned patients. A logistic regression model was created in regards to demographic and hospitalization data and can be found in Appendix A. Although statistically significant (*p* < 0.0001), the model accounted for only a limited portion of variability (R^2^ = 0.0123), with the following statistically significant observations: each additional hospital day increased the risk of abandonment by 2.21%; premature infants were 14.97 times more likely to be abandoned than non-premature infants; and malnourished children had the highest risk, being 46.02 times more likely to experience abandonment.

Data relating to antibiotic class resistance in abandoned patients can be seen in Table 9. The following statistically significant differences were observed: a higher resistance in abandoned patients for fluoroquinolones, aminoglycosides, and reserve antibiotics for 2019 and for cephalosporins and urinary antibiotics in 2023.

Lastly, a second logistic model was run in regards to antibiotic classes and resistance phenotypes, for which the results are observable in Table 10. The model was considered statistically significant (*p* = 0.0119); however, it had a limited portion of variability explained (R^2^ = 0.0031). Key factors were resistance to cephalosporins (*p* = 0.0304) and reserve antibiotics (*p* = 0.0049). In terms of the odds ratio, this means that patients with resistance to cephalosporins were 5.17 times more likely to experience abandonment, while those with resistance towards reserve antibiotics were 5.64 times more likely. Of note is that among the resistance phenotypes, the most important one was ESBL, although it was borderline not statistically significant (*p* = 0.0536).

## 3. Discussion

The COVID-19 pandemic disrupted healthcare services, including pediatric settings, where delays in routine care and increased antibiotic use may have influenced bacterial infection trends and resistance patterns. Prolonged hospital stays, invasive procedures, and empirical antibiotic use created an environment conducive to the emergence of MDR bacterial and fungal strains.

### 3.1. Demographics and Bacterial Trends

While sex distribution remained relatively stable, a notable shift in the geographic distribution of patients was observed, with urban predominance in 2019 and 2021 but equalization between urban and rural origins by 2023. Infants consistently represented the most affected age group, aligning with previous studies that reported higher hospitalization rates among neonates and young children during the COVID-19 pandemic [24].

Hospital admissions fluctuated significantly, with a 50% decrease during the pandemic year (2021) compared to pre-pandemic levels, likely due to parental hesitancy, lockdowns, and restrictions on non-emergency cases [25]. During the pandemic, hospital resource utilization was markedly different, with an increase in ICU, Surgery, and Dialysis admissions compared to pre- and post-pandemic years. This trend suggests that only severe cases were hospitalized during 2021, while milder infections were managed in outpatient settings [15,25]. Post-pandemic, hospitalization patterns reverted to pre-pandemic values, as observed in national and international trends [25,26,27,28,29].

The median LoS increased from 3 days (2019) to 5 days (2021), returning to 2 days in 2023, likely due to stricter admission criteria and greater disease severity during the pandemic [16,30,31]. Notably, ICU/NICU stays were longest in 2021 (median: 10 days), consistent with findings that critically ill pediatric patients had prolonged hospitalizations during the COVID-19 wave.

Sample collection patterns shifted according to hospital resource utilization. Nasal secretions were least collected during the pandemic but increased post-pandemic, likely due to a resurgence of viral–bacterial co-infections. Wound secretions, catheter samples, and hypopharyngeal aspirates peaked in 2021 and 2023, suggesting increased nosocomial infections or monitoring efforts. Blood cultures and conjunctival secretions were more frequently sampled post-pandemic, reflecting expanded diagnostic protocols. In contrast, CSF and pleural fluid sample collection declined over time, likely due to fewer invasive infections in pediatric patients during and after the pandemic [32].

Throughout all years, Gram-negative bacilli dominated infections, with *Escherichia coli* (16.99%), *Klebsiella pneumoniae* (8.63%), and *Pseudomonas aeruginosa* (8.83%) being among the most frequently identified pathogens. Among Gram-positive bacteria, *Staphylococcus aureus* (13.96%) and group A *Streptococcus* (8.51%) were most prevalent. These trends are consistent with previous studies reporting a sustained presence of resistant *Klebsiella* and *Escherichia coli* infections in Europe, particularly in Romania, where carbapenem-resistant *Klebsiella pneumoniae* was already a major concern before COVID-19 [33,34,35].

Gram-negative pathogens remained dominant, with *Escherichia coli*, *Klebsiella pneumoniae*, and *Pseudomonas aeruginosa* consistently among the most frequent isolates. Gram-positive infections fluctuated, with *Staphylococcus aureus* peaking during the pandemic before declining post-pandemic, while Group A *Streptococcus* surged in 2023. Fungal infections were less common overall (<10%), though *Candida* spp. remained a concern in high-risk pediatric patients. Respiratory infections declined during the pandemic but rebounded post-pandemic, with pneumococcal and streptococcal infections returning to pre-pandemic levels. These trends were similar to international trends [16,28,33,36].

### 3.2. Antimicrobial Testing and Resistance Patterns

The most frequently tested antimicrobial classes were fluoroquinolones (93.07%), aminoglycosides (90.04%), and penicillins (88.40%), demonstrating the heavy reliance on these antibiotics in pediatric settings. The post-pandemic period showed an increase in cephalosporin and carbapenem usage, consistent with the observed surge in hospital-acquired infections with carbapenem-resistant *Klebsiella pneumoniae* as reported by Luo and Chen [37]. Also, in a similar trend to Miao et al., who although focused on pediatric Mycoplasma pneumoniae infections post-pandemic, a similar pattern was observed relating to increased resistance in fluoroquinolone and macrolide use in 2023 [38]. In turn, this suggests an increased reliance on these antibiotic classes for respiratory infections, which were previously suppressed due to COVID-19-related lockdowns and reduced viral co-infections.

During the pandemic, there was a significant decline in combination therapy and reserve antibiotic usage (IRR = 0.82, *p* = 0.0076 and IRR = 0.75, *p* = 0.0001, respectively), suggesting a shift toward monotherapy or reduced availability of certain antibiotics. Cephalosporins also showed a decreasing trend (IRR = 0.87, *p* = 0.0520), similar to findings from Silva-Costa et al., where pediatric invasive pneumococcal disease cases initially declined before rebounding post-pandemic [39]. Meanwhile, glycopeptides and antifungal polyenes were more frequently used in 2019 than in 2021 (IRR = 7.8, *p* < 0.0001 and IRR = 3.93, *p* = 0.0278, respectively), reflecting a higher reliance on broad-spectrum antimicrobials prior to the pandemic, a trend also noted in Wu et al. [40]. However, as antibiotic stewardship efforts evolved, pandemic-driven shifts favored the increased use of fluoroquinolones, carbapenems, and aminoglycosides [41].

Post-pandemic, cephalosporins and carbapenems saw significantly increased testing and usage compared to 2019 (IRR = 1.21, *p* = 0.0032 and IRR = 1.52, *p* < 0.0001, respectively), aligning with the global resurgence of MDR *Klebsiella pneumoniae* and *Pseudomonas aeruginosa* [37,42]. In contrast, fluoroquinolone, aminoglycoside, and lincosamide use declined post-pandemic (IRR = 0.82, *p* = 0.0012; IRR = 1.38, *p* < 0.0001; IRR = 1.54, *p* = 0.0001, respectively). This trend was also observed in antibiotic stewardship studies, which reported improved prescribing practices after the peak of COVID-19-related overuse [41,42].

By 2023, fluoroquinolone, urinary antibiotic, and antifungal usage increased significantly compared to 2021 (IRR = 1.27, *p* = 0.0010; IRR = 2.43, *p* < 0.0001; IRR = 4.84, *p* < 0.0001, respectively), mirroring the post-pandemic rebound in pediatric infections reported by the international literature [42]. The rise in urinary antibiotic use could be linked to increased pediatric urinary tract infections (UTIs), similar to trends found in post-pandemic surveillance data [33,39]. Conversely, cephalosporin, carbapenem, aminoglycoside, macrolide, lincosamide, and reserve antibiotic usage declined in 2023 compared to 2021 (*p*-values < 0.05 for all categories). This suggests a gradual normalization of antimicrobial prescribing and is in line with efforts to curb excessive broad-spectrum antibiotic use post-pandemic, as noted by Mohammed et al. [41].

Overall, 69.92% of tested samples exhibited resistance to at least one antimicrobial class, with higher resistance rates in 2019 (72.69%) compared to 2021 (67.05%) and 2023 (69.16%). A similar decrease in AMR during the pandemic was reported by Silva-Costa et al., likely due to reduced hospitalizations and lower infection transmission [39].

Regarding phenotypes of resistance, both MRSA and MRCoNS peaked during the pandemic, with a significant decrease afterwards. This trend has been also described by Golli et al. [42]. On the contrary, VRE did not show statistically significant changes across the three timeframes, which is consistent with Zhang et al., who also found limited pandemic impact on VRE rates in pediatric settings [43].

ESBL producers, particularly *Escherichia coli* (32.96%), *Klebsiella pneumoniae* (28.72%), and *Pseudomonas aeruginosa* (13.53%), followed a declining trend between 2019 and 2021 before rebounding in 2023. The overall ESBL rate was significantly lower in 2021 than in 2023 (IRR = 0.64, *p* < 0.0001), indicating a temporary pandemic-related decline in broad-spectrum beta-lactamase-producing pathogens. This trend supports Fallah et al., which documented reduced transmission of ESBL bacteria during the pandemic due to lower pediatric hospitalizations [44]. However, ESBL-producing *Serratia marcescens* and *Enterobacter* spp. showed significant post-pandemic increases (IRR = 3.35, *p* = 0.0006 and IRR = 2.27, *p* < 0.0001, respectively), mirroring findings from AlFonaisan et al., which linked pandemic-era antimicrobial misuse to increased beta-lactamase resistance in opportunistic pathogens [45].

Carbapenemase-resistant organisms peaked during the pandemic before stabilizing post-pandemic. *Pseudomonas aeruginosa* (49.14%), *Klebsiella pneumoniae* (22.57%), and *Chryseobacterium* spp. (6.00%) were the dominant CRO species. *Pseudomonas aeruginosa* showed a significant decline post-pandemic (IRR = 0.34, *p* < 0.0001), aligning with findings from Golli et al., which reported a post-COVID-19 reduction in carbapenem-resistant *Pseudomonas aeruginosa* infections due to improved infection control measures [42]. *Klebsiella pneumoniae* followed a different pattern, increasing significantly in 2023 compared to pre-pandemic levels (IRR = 1.35, *p* = 0.0283), consistent with trends in pediatric ICUs [44]. *Enterobacter* spp. demonstrated a significant pandemic peak (IRR = 0.29, *p* = 0.0161) before declining post-pandemic, similar to findings from Zhang et al., which attributed this trend to pandemic-driven antimicrobial selection pressure [43].

MDR bacteria were prevalent across all study years, with *Escherichia coli* (19.91%), *Staphylococcus aureus* (13.82%), and *Klebsiella pneumoniae* (13.82%) being the most frequently isolated MDR pathogens. The overall MDR incidence dropped during the pandemic (IRR = 1.09, *p* = 0.0001) before returning to pre-pandemic levels in 2023. This trend was also observed by Kaushik et al., who linked reduced pediatric hospital admissions in 2021 to fewer MDR infections [46]. *Pseudomonas aeruginosa* showed a decreasing MDR trend between 2019 and 2023 (IRR = 1.11, *p* = 0.0256), supporting findings from AlFonaisan et al., which documented a similar MDR *Pseudomonas* decline [45]. Conversely, *Enterobacter* spp. exhibited an increasing MDR trend post-pandemic (IRR = 1.65, *p* = 0.0009), consistent with reports from Fallah et al., which identified post-pandemic MDR increases among pediatric *Enterobacter* infections [44].

For XDR bacteria, rates peaked in 2019 before declining during and after the pandemic (IRR = 1.47, *p* = 0.0006). However, *Klebsiella pneumoniae* showed an increase in 2023 (IRR = 1.74, *p* = 0.0018), reflecting a post-pandemic resurgence in extensively resistant strains [46,47].

Additionally, attention should be given to neonates born to mothers with vaginal dysbiosis or infections, as these infants are at a higher risk of admission to NICUs and preterm wards. Such vulnerable neonates often exhibit increased susceptibility to opportunistic infections, including fungal infections caused by *Candida* spp. and MDR Gram-negative bacteria like *Klebsiella pneumoniae* and *Pseudomonas* spp. Also, a key strategy to reduce antibiotic-resistant infections in pediatric populations involves optimizing antibiotic duration, particularly for common infections such as UTIs. Recent findings suggest that short-course antibiotic therapy (3–5 days) is non-inferior to standard 7–10-day regimens for uncomplicated pediatric UTIs while significantly lowering the risk of antimicrobial resistance and reinfection [48]. This is particularly relevant in the context of our findings, where *Escherichia coli* and *Pseudomonas* spp. were among the most frequently isolated pathogens. Individualized treatment plans with prolonged or targeted therapy may still be warranted, and resistance surveillance becomes even more critical. Moreover, the post-pandemic rebound in MDR and XDR strains observed in our study reinforces the need for hospital-specific antimicrobial stewardship programs that integrate local resistance trends into empiric therapy guidelines.

Given these risks, a multidisciplinary approach involving early maternal screening, improved neonatal infection prevention, and targeted antimicrobial management is critical to reducing hospital-acquired infections and AMR burden in neonatal care settings [49,50,51,52].

### 3.3. Interdisciplinary Research—Pediatric Abandoment in the Hospital

Pediatric hospital abandonment remains an underrecognized but critical issue that intertwines healthcare, socioeconomic disparities, and public health infrastructure. The data presented in the present study highlight notable trends in pediatric abandonment from 2019 through 2023, emphasizing an overall decline in national cases but an increasing percentage of cases in the Western region and Timis County. These findings align with broader global trends that have linked economic hardship, family disruptions, and limited access to social support systems with rising abandonment rates, particularly during and after the COVID-19 pandemic [17]. Although total cases are declining, specific regions may be experiencing exacerbated socioeconomic challenges that contribute to pediatric abandonment. Studies from other countries have found similar patterns, with regional disparities influenced by economic downturns, unemployment, and healthcare access limitations during and after the pandemic [53,54,55]. At the hospital level, abandonment cases showed an important dip during 2021 (0.52%) before rising again in 2023 (1.69%), possibly reflecting temporary pandemic-induced barriers, such as restricted hospital access and lower pediatric admissions.

The logistic regression analysis indicated that prolonged hospitalization increased the likelihood of abandonment by 2.21% per additional hospital day, highlighting the role of medical complexity and prolonged care needs in driving family separation. Prematurity emerged as a significant factor, with premature infants being 14.97 times more likely to be abandoned, while malnourished children had the highest risk at 46.02 times greater odds. These findings are consistent with previous studies showing that children with significant medical needs or chronic conditions are at the highest risk of abandonment due to the perceived burden on caregivers [54,56].

Among abandoned pediatric patients at “Louis Turcanu” Children’s Emergency Hospital, infection rates varied across the years but showed notable spikes in 2023 (1.02%). A statistically significant association was found with *Stenotrophomonas maltophilia*, an emerging MDR pathogen. This supports previous research suggesting that abandoned children are at a higher risk of nosocomial infections due to prolonged hospital stays [54].

Additionally, antibiotic resistance patterns in abandoned patients reveal concerning trends. In 2019, higher resistance was observed for fluoroquinolones, aminoglycosides, and reserve antibiotics, while in 2023, cephalosporins and urinary antibiotics exhibited significantly higher resistance rates. The regression model further confirmed that resistance to cephalosporins (OR = 5.17) and reserve antibiotics (OR = 5.64) were strong predictors of pediatric abandonment cases. These findings emphasize that abandoned children not only face heightened infection risks but also tend to harbor more resistant pathogens, likely due to prolonged hospital stays and extensive antibiotic exposure. Studies in this regard are scarce, as a main focus in regards to pediatric abandonment in healthcare units relates to HIV infections rather than bacterial or fungal [56,57].

### 3.4. Limitations

As with all retrospective studies, it is important to note that this paper has several limitations, including some inherent to this type of study. Although a thorough examination of the dataset was employed, the results may not be representative for the general pediatric population in Romania or Europe. Also, in order to have a complete dataset, some patients were not included in the study due to missing information. Regarding the pandemic, infection with SARS-CoV-2 was not recorded and analyzed, as the subject of co-/superinfection has been a major focus of research in recent times. However, it can still be viewed as a potential confounding variable and, as such, is declared here.

Another important limitation may relate to the switch of AST change in guidelines for the disk diffusion method. For the years 2019 and 2021, the hospital followed the CLSI guidelines, while starting in 2023, the EUCAST standards were employed.

Regarding the studied antimicrobial classes and resistance phenotypes, an analysis was performed for the whole hospital irrespective of the origin of the sample. It is possible that certain wards or ward types (general hospitalization, admission to intensive care units, outpatients) may have observed different rates than the overall trends.

As this study is monocentric, it may not fully represent national trends. The inclusion of national and regional data (e.g., for abandonment cases) helps broaden context, but the core infection and resistance analysis is hospital specific. Another important limitation for the secondary objective of abandonment implication relates to the limited sample of abandoned patients itself.

The authors acknowledge the need for further research, particularly in the form of a meta-analysis on a country-wide level, which would offer a more systematic and quantitative synthesis of the national situation. A meta-analysis would improve data accuracy by addressing inconsistencies in study design and reporting, ultimately allowing for more precise estimates of bacterial prevalence and resistance trends.

Lastly, the availability of the recent literature on pediatric hospital abandonment, particularly in the context of post-pandemic healthcare in Eastern Europe, appears to be scarce. While the provided references used to contextualize this issue are older, they remain among the few empirical sources addressing institutional abandonment of infants in maternity and pediatric wards. This underscores both a literature gap and the need for updated, systematic research on child protection failures in hospital settings—especially in regions with persistent healthcare and social service inequalities.

## 4. Materials and Methods

### 4.1. Study Design

This study was a retrospective observational analysis conducted at the Children’s Emergency Hospital “Louis Țurcanu” Timișoara, focusing on AMR patterns in pediatric bacterial and fungal infections. The study analyzed changes over three distinct time periods, 2019 pre-pandemic, 2021 pandemic, and 2023 post-pandemic, in order to evaluate the impact of the COVID-19 pandemic on pediatric AMR patterns and hospital abandonment rates. Data were extracted from hospital medical records and institutional databases, ensuring a comprehensive assessment of the selected variables. Inclusion criteria were pediatric patients (less than 18 years) with a complete demographic and infectious dataset.

### 4.2. Selection of Data

The following demographic information were extracted: year of admission, abandonment status, patient age, sex, location of residence, hospital ward, and length of stay. Data related to pediatric microbial infection included the sample type, identified pathogen, antibiotics used for susceptibility testing (Appendix A), and resistance profiles, as defined by Magiorakos et al. [20]. Also, the following common specific phenotypes were noted: MRSA (methicillin-resistant *Staphylococcus aureus*) and MRCoNS (methicillin-resistant coagulase negative Staphylococci), VRE (vancomycin-resistant Enterococci), ESBL (extended-spectrum beta-lactamase producers), CRO (carbapenem-resistant organisms), and CRE (carbapenem-resistant Entrobacterales) [21,22,23].

For microbiological analysis, clinical samples were collected from pediatric patients with suspected bacterial or fungal infections based on clinical signs and physician referral. Sample types included urine, nasal secretion, wound secretion, hypopharyngeal aspirate, pharyngeal exudate, blood, stool, skin, catheter tips, cerebrospinal fluid (CSF), pleural fluid, peritoneal fluid, umbilical secretion, conjunctival secretion, otic secretion, vaginal secretion, oral lesions, and male genital secretions. A small number of unclassified samples were labeled as “other”. Sampling followed strict aseptic protocols as per hospital infection control guidelines. After collection, samples were transported promptly to the microbiology lab in sterile containers, under cold chain when necessary, and processed within 2 h. Cultures were incubated on appropriate media under standard atmospheric conditions. Identification was confirmed through cultural, morphological, and biochemical characteristics. If doubt persisted, matrix-assisted laser desorption/ionization time-of-flight mass spectrometry was employed. Depending on the ward origin, certain isolates were identified and tested using the Vitek 2 automated system (BioMerieux. Marcy-l’Étoile, France)

Regarding the methodology of antimicrobial susceptibility testing, testing was performed according to hospital protocols using the disk diffusion method on Muller–Hinton agar (Biomaxima Romania, Cluj-Napoca, Romania) after a standard inoculum adjusted to 0.5 McFarland was swabbed on its surface using the ready-made antibiotics’ supplied dispenser. For the years 2019 and 2021, the hospital followed the Clinical Laboratory Standards Institute (CLSI) guidelines, while starting with 2023, the European Committee on Antimicrobial Susceptibility Testing (EUCAST) standards were employed.

### 4.3. Child Abandonment Analysis

For the purposes of this study, abandonment is operationally defined as cases in which a pediatric patient remained hospitalized beyond medical necessity due to the absence of a legal guardian upon discharge. The following data were also taken into consideration: total number of abandonment cases per year and rate at regional and county levels.

### 4.4. Statistical Analysis

The recorded data were compiled into a spreadsheet file. The minimum sample size was determined using G*Power software (version 3.1.9.7, Heinrich Heine Unviersitat, Dusseldorf, Germany) through an a priori test, calculating the required sample for a small effect size (0.1) with a power of 95%. The analysis indicated a minimum of 1548 patients. Subsequently, data analysis was conducted using MedCalc Statistical Software (version 20.218, MedCalc Software bv, Ostend, Belgium).

Data normality was assessed via the Shapiro–Wilk test. Since the continuous variables did not follow a normal distribution, the Mann–Whitney test or the Kruskal–Wallis test with Dunn’s post hoc analysis was applied to identify statistically significant differences. For categorical variables, contingency tables were constructed, and the Chi-square test was used. Additionally, incidence rate ratio (IRR) analysis was performed to compare infection rates and abandonment trends, providing the 95% Confidence Interval (95% CI). The rates were compared as follows: 2019/2021, 2019/2023, and 2023/2021. Regarding interpretation of IRR analysis, a rapport close to 1 was considered similar incidence between the two years, a rapport >1 meant that the first year displayed a higher rate than the second, and a rapport <1 meant that the second year displayed a higher incidence.

Logistic regression modeling was employed to identify predictors of hospital abandonment, incorporating independent variables such as age, sex, place of residence, hospital stay duration, and admission to intensive care services. Also, phenotype resistance was studied using this test.

### 4.5. Ethics

This study was conducted in full compliance with the ethical standards outlined in the Declaration of Helsinki (2013 revision) and with national regulations governing research involving human subjects. Ethical approval was obtained from the Ethics Committee of the “Victor Babes” University of Medicine and Pharmacy, Timisoara, Romania, which oversees research activities involving the “Louis Țurcanu” Children’s Emergency Hospital. The study was approved under protocol number 31/29.04.2024.

As the study involved retrospective analysis of microbiological and demographic data extracted from hospital records, no direct patient contact or intervention occurred, and no identifiable personal data (e.g., name, address, or personal identification number) were recorded, stored, or analyzed. Data were anonymized prior to analysis to ensure patient confidentiality. Informed consent for microbiological testing was obtained at the time of hospital admission or sample collection as part of routine medical care in accordance with Romanian national healthcare regulations.

## 5. Conclusions

This study highlights the long-term impact of the COVID-19 pandemic on pediatric infections, AMR, and hospital abandonment trends. While initial reductions in resistance rates were observed due to lower hospital admissions during the pandemic, post-pandemic resurgence of MDR and XDR pathogens—including MRSA, VRE, ESBL, and CRO—raises critical concerns for pediatric healthcare. The widespread use of broad-spectrum antibiotics during the pandemic has likely accelerated AMR trends, emphasizing the urgent need for robust antimicrobial stewardship programs.

Beyond microbial resistance, this study also underscores the intersection between pediatric abandonment and infectious disease risks. Abandoned children, particularly those requiring prolonged hospital stays due to prematurity, malnutrition, or chronic conditions, are disproportionately affected by healthcare-associated infections and multidrug-resistant pathogens. The increasing regional burden of pediatric abandonment, despite a national decline, suggests persistent socioeconomic challenges that demand a multidimensional response.

Moving forward, continued AMR surveillance, optimized infection control practices, and stricter antimicrobial stewardship will be essential to mitigate the long-term consequences of pandemic-era prescribing practices. Furthermore, integrating social interventions, predictive screening for high-risk neonates, and enhanced hospital hygiene protocols will be critical in reducing AMR burden and improving pediatric health outcomes. As healthcare systems recover from the disruptions of COVID-19, a comprehensive, interdisciplinary approach is necessary to address the dual challenge of rising resistance and increasing vulnerability among abandoned pediatric patients.

## Figures and Tables

**Table 1 antibiotics-14-00411-t001:** Tested antimicrobial Chi^2^ test results.

Class	Antibiotic	Chi^2^ Statistic	*p*	Notes
**Penicillins**		**145.14**	**<0.0001**	
	Amoxicillin	NA	NA	Not tested in 2019 and 2023
	Ampicillin	42.1	<0.0001	
	Oxacillin	67.33	<0.0001	
	Penicillin	17.33	0.0017	
	Piperacillin	64.34	<0.0001	
	Methicillin	33.87	<0.0001	
	Ticarcillin	64.5	<0.0001	
**Cephalosporins**		**263.34**	**<0.0001**	
	Cefepime	47.31	<0.0001	
	Cefotaxime	127.85	<0.0001	
	Cefoxitin	95.17	<0.0001	
	Ceftazidime	55.07	<0.0001	
	Ceftibuten	123.83	<0.0001	
	Ceftizoxime	NA	NA	Not tested in 2021 and 2023
	Ceftriaxone	45.71	<0.0001	
	Cefuroxime	NA	NA	Not tested in 2019 and 2021
**Combination therapy (penicillins and cephalosporins)**		**165.33**	**<0.0001**	
	Amoxicillin/Clavulanic Acid	116.48	<0.0001	
	Ampicillin/Sulbactam	12.53	0.0138	
	Ceftazidime/Avibactam	561.48	<0.0001	Not tested in 2019
	Piperacillin/Tazobactam	12.97	0.0114	
	Ticarcillin/Clavulanic Acid	91.79	<0.0001	
**Carbapenems**		**66.63**	**<0.0001**	
	Ertapenem	143.86	<0.0001	
	Imipenem	94.04	<0.0001	
	Meropenem	11.29	0.0235	
**Fluoroquinolones**		**188.49**	**<0.0001**	
	Ciprofloxacin	163.38	<0.0001	
	Levofloxacin	333.86	<0.0001	
	Moxifloxacin	26.02	<0.0001	
	Norfloxacin	92.44	<0.0001	
	Ofloxacin	115.31	<0.0001	Not tested in 2021
**Aminoglycosides**		**63.41**	**<0.0001**	
	Amikacin	40.64	<0.0001	
	Gentamicin	162.63	<0.0001	
	Gentamicin-HL	44.23	<0.0001	
	Kanamycin	35.85	<0.0001	
	Neomycin	NA	NA	Not tested in 2019 and 221
	Netilmicin	4.55	0.3366 *	
	Streptomycin-HL	142.05	<0.0001	
	Tobramycin	22.89	0.0001	
**Macrolides**		**127.91**	**<0.0001**	
	Azithromycin	60.17	<0.0001	
	Clarithromycin	NA	NA	Not tested in 2019 and 2023
	Erythromycin	101.59	<0.0001	
**Lincosamides**		**54.84**	**<0.0001**	
	Clindamycin	83.38	<0.0001	
	Clindamycin- inducible	6.77	0.0339	
**Cyclines**		**44.29**	**<0.0001**	
	Minocycline	0.73	0.6935 *	Not tested in 2023
	Tetracycline	104.44	<0.0001	
	Tigecycline	34.27	<0.0001	
**Glycopeptides**		**90.95**	**<0.0001**	
	Teicoplanin	108.31	<0.0001	
	Vancomycin	2.51	0.6429 *	
**Urinary**		**198.79**	**<0.0001**	
	Nalidixic Acid	1.36	0.8504 *	
	Fosfomycin	218.63	<0.0001	
	Nitrofurantoin	85.35	<0.0001	
**Other**		**234.92**	**<0.0001**	
	Aztreonam	47.31	<0.0001	
	Chloramphenicol	36.98	<0.0001	
	Fusidic Acid	73.29	<0.0001	
	Rifampicin	58.54	<0.0001	
**Reserve**		**119.21**	**<0.0001**	
	Colistin	347.36	<0.0001	
	Linezolid	124.86	<0.0001	
	Cotrimoxazole	52.94	<0.0001	
**Antifungals—Polyenes**		**89.73**	**<0.0001**	
	Amphotericin B	8.95	0.0624 *	
	Nystatin	111.84	<0.0001	
**Antifungals—Azoles**		**288.73**	**<0.0001**	
	Clotrimazole	NA	NA	Not tested in 2021 and 2023
	Econazole	99.96	<0.0001	
	Fluconazole	8.02	0.0908	
	Itraconazole	378.15	<0.0001	
	Ketoconazole	15.81	0.0033	
	Miconazole	107.2	<0.0001	
	Voriconazole	NA	NA	Not tested in 2021 and 2023
**Antifungals—Other**		**65.84**	**<0.0001**	
	Caspofungin	0.768	0.0882	Not tested in 2021
	Flucytosine	67.02	<0.0001	

NA: not applicable, HL: high level, *: failed to reach statistical significance.

**Table 2 antibiotics-14-00411-t002:** Rate comparison for antimicrobial classes.

IRR, (95%CI; *p*)	2019 vs. 2021	2019 vs. 2023	2023 vs. 2021
Penicillins	0.94 (0.83–1.07; 0.3453)	0.96 (0.87–1.07; 0.5007)	0.98 (0.87–1.10; 0.7172)
Cephalosporins	0.87 (0.76–1.00; 0.0520)	1.21 (1.07–1.38; 0.0032) *	0.72 (0.62–0.83; <0.0001) *
Combination therapy	0.82 (0.71–0.95; 0.0076) *	0.97 (0.85–1.10; 0.6023)	0.85 (0.74–0.98; 0.0257) *
Carbapenems	1.13 (0.92–1.40; 0.2511)	1.52 (1.25–1.85; <0.0001) *	0.74 (0.59–0.93; 0.0096) *
Fluoroquinolones	1.04 (0.90–1.20; 0.6167)	0.82 (0.73–0.93; 0.0012) *	1.27 (1.10–1.46; 0.0010) *
Aminoglycosides	1.09 (0.95–1.25; 0.2309)	1.38 (1.22–1.57; <0.0001) *	0.79 (0.68–0.91; 0.0014) *
Macrolides	0.89 (0.74–1.07; 0.2199)	1.13 (0.95–1.34; 0.1699)	0.79 (0.65–0.95; 0.0142) *
Lincosamides	1.09 (0.87–1.37; 0.4343)	1.54 (1.25–1.90; 0.0001) *	0.71 (0.56–0.90; 0.0052) *
Cyclines	0.79 (0.63–0.99; 0.0419)	1.16 (0.94–1.43; 0.1780)	0.69 (0.55–0.86; 0.0011) *
Glycopeptides	7.8 (3.94–15.44; <0.0001) *	5.91 (3.66–9.55; <0.0001) *	1.32 (0.60–2.91; 0.4917)
Urinary	1.25 (0.99–1.58; 0.0640) *	0.51 (0.43–0.61; <0.0001) *	2.43 (1.96–3.01; <0.0001) *
Reserve	0.75 (0.65–0.87; 0.0001) *	0.91 (0.80–1.04; 0.1885)	0.82 (0.71–0.95; 0.0077) *
Other antibiotics	1 (0.76–1.33; 0.9945)	0.29 (0.24–0.36; <0.0001) *	3.44 (2.70–4.39; <0.0001) *
Polyenes	3.93 (1.16–13.32; 0.0278) *	0.15 (0.09–0.24; <0.0001) *	26.32 (8.36–82.86; <0.0001) *
Azoles	5 (3.56–7.03; <0.0001) *	1.03 (0.87–1.22; 0.7076)	4.84 (3.44–6.81; <0.0001) *
Other antifungals	8.61 (3.98–18.60; <0.0001) *	0.74 (0.56–0.96; 0.0243) *	11.71 (5.46–25.10; <0.0001) *

*: statistically significant.

**Table 3 antibiotics-14-00411-t003:** Rate comparison for specific phenotypes: MRSA, MRCoNS, and VRE.

IRR (95%CI; *p*)	2019 vs. 2021	2019 vs. 2023	2023 vs. 2021
MRSA	0.42 (0.29–0.59; <0.0001) *	1.39 (0.90–2.16; 0.1172)	0.30 (0.20–0.45; <0.0001) *
MRCoNS	0.63 (0.43–0.93; 0.0159) *	1.57 (1.08–2.26; 0.0121) *	0.40 (0.28–0.59; <0.0001) *
VRE	0.40 (0.03–2.81; 0.3144)	0.46 (0.05–2.96; 0.5076)	0.71 (0.18–3.33; 0.5886)

*: statistically significant.

**Table 4 antibiotics-14-00411-t004:** Rate comparison for ESBL bacteria.

IRR (95%CI; *p*)	2019 vs. 2021	2019 vs. 2023	2023 vs. 2021
Overall ESBL	1.00 (0.89–1.12; 0.9414)	1.54 (1.37–1.74; <0.0001) *	0.64 (0.57–0.73; <0.0001) *
*Escherichia coli*	1.25 (1.04–1.51; 0.0165) *	1.36 (1.15–1.62; 0.0004) *	0.92 (0.75–1.13; 0.4222)
*Klebsiella pneumoniae*	1.12 (0.95–1.33; 0.1745)	1.03 (0.90–1.18; 0.6692)	1.09 (0.92–1.29; 0.3160)
*Pseudomonas aeruginosa*	0.80 (0.60–1.05; 0.1130)	1.24 (0.90–1.72; 0.1907)	0.64 (0.47–0.88; 0.0060) *
*Serratia marcescens*	1.04 (0.90–1.20; 0.5650)	3.35 (1.68–6.65; 0.0006) *	0.31 (0.16–0.62; 0.0010) *
*Enterobacter* spp.	0.93 (0.83–1.03; 0.1577)	2.27 (1.53–3.35; <0.0001) *	0.41 (0.28–0.60; <0.0001) *
*Proteus mirabilis*	2.29 (0.56–9.38; 0.2479)	0.72 (0.38–1.34; 0.2974)	3.20 (0.79–12.95; 0.1029)
*Klebsiella* spp.	0.94 (0.42–2.12; 0.8871)	1.41 (0.59–3.40; 0.4379)	0.67 (0.24–1.81; 0.4275)
*Morganella* spp.	1.50 (0.67–3.34; 0.3206)	1.17 (0.86–1.58; 0.3178)	1.29 (0.55–3.02; 0.5647)
*Chryseobacterium* spp.	0.84 (0.57–1.22; 0.3585)	NA	NA
*Citrobacter* spp.	0.70 (0.35–1.41; 0.3199)	1.30 (0.55–3.09; 0.5519)	0.54 (0.23–1.24; 0.1448)
*Acinetobacter baumannii*	3.88 (0.50–30.10; 0.1942)	6.88 (0.87–54.35; 0.0673)	0.56 (0.04–8.58; 0.6802)
Other	3.40 (0.42–27.59; 0.2520)	NA	NA
*Sphingomonas paucimobilis*	0.22 (0.03–1.49; 0.1217)	NA	NA
*Pseudomonas* spp.	0.50 (0.06–3.91; 0.5087)	1.00 (0.13–7.57; 0.9453)	0.50 (0.11–2.19; 0.3572)
*Serratia* spp.	3.00 (0.31–28.84; 0.3414)	NA	NA
*Acinetobacter* spp.	NA	NA	0.27 (0.03–2.51; 0.2479)

*: statistically significant, NA: not applicable.

**Table 5 antibiotics-14-00411-t005:** Rate comparison for CRO bacteria.

IRR (95%CI; *p*)	2019 vs. 2021	2019 vs. 2023	2023 vs. 2021
Overall CRO	0.51 (0.40–0.65; <0.0001) *	1.35 (1.03–1.78; 0.0283) *	0.38 (0.29–0.49; <0.0001) *
*Pseudomonas aeruginosa*	0.58 (0.44–0.76; 0.0001) *	1.70 (1.14–2.53; 0.0088) *	0.34 (0.24–0.49; <0.0001) *
*Klebsiella pneumoniae*	0.42 (0.22–0.80; 0.0086) *	0.31 (0.17–0.55; 0.0001) *	1.35 (0.83–2.19; 0.2203)
*Proteus mirabilis*	2.45 (0.32–18.67; 0.3880)	2.04 (0.56–7.39; 0.2779)	1.20 (0.13–11.02; 0.8719)
*Chryseobacterium* spp.	0.78 (0.55–1.10; 0.1584)	NA	NA
*Acinetobacter baumannii*	3.24 (0.40–25.87; 0.2683)	5.74 (0.70–46.71; 0.1026)	0.56 (0.04–8.58; 0.6802)
*Escherichia coli*	1.89 (0.37–9.66; 0.4466)	1.78 (0.43–7.38; 0.4300)	1.06 (0.18–6.32; 0.9471)
*Sphingomonas paucimobilis*	0.56 (0.12–2.68; 0.4640)	NA	NA
*Enterobacter* spp.	0.29 (0.10–0.79; 0.0161) *	3.72 (0.43–31.90; 0.2307)	0.08 (0.01–0.55; 0.0110) *
*Klebsiella* spp.	0.86 (0.09–8.58; 0.8956)	1.29 (0.13–13.17; 0.8323)	0.67 (0.05–9.66; 0.7663)
Other	1.70 (0.17–17.16; 0.6528)	0.90 (0.09–8.69; 0.9274)	1.89 (0.13–26.78; 0.6383)
*Morganella* spp.	NA	NA	NA
*Pseudomonas* spp.	NA	3.00 (0.23–39.61; 0.4040)	NA
*Acinetobacter* spp.	NA	NA	1.60 (0.20–12.99; 0.6600)
*Proteus* spp.	NA	NA	NA
*Serratia marcescens*	NA	NA	0.91 (0.18–4.57; 0.9080)
*Serratia* spp.	NA	NA	NA

*: statistically significant, NA: not applicable.

**Table 6 antibiotics-14-00411-t006:** Rate comparison for all MDR organisms.

IRR (95%CI; *p*)	2019 vs. 2021	2019 vs. 2023	2023 vs. 2021
Overall MDR	1.09 (1.05–1.14; 0.0001) *	1.03 (1.00–1.07; 0.0747)	1.06 (1.01–1.11; 0.0119) *
*Escherichia coli*	0.99 (0.92–1.07; 0.8458)	0.93 (0.87–0.99; 0.0246) *	1.07 (1.00–1.15; 0.0623)
*Klebsiella pneumoniae*	1.00 (0.98–1.03; 0.9062)	1.00 (0.98–1.03; 0.6991)	1.00 (0.97–1.02; 0.8265)
*Staphylococcus aureus*	1.10 (0.95–1.28; 0.1945)	0.65 (0.59–0.71; <0.0001) *	1.71 (1.51–1.93; <0.0001) *
*Pseudomonas aeruginosa*	1.27 (1.13–1.41; <0.0001) *	1.11 (1.01–1.21; 0.0256) *	1.14 (1.01–1.29; 0.0330) *
*Streptococcus pneumoniae*	0.96 (0.84–1.10; 0.5633)	1.24 (1.08–1.42; 0.0017) *	0.77 (0.66–0.91; 0.0016) *
CoNS	1.11 (1.03–1.20; 0.0082) *	1.03 (1.01–1.06; 0.0143) *	1.08 (0.99–1.17; 0.0806)
*Proteus mirabilis*	1.00 (0.92–1.09; 0.9409)	1.05 (0.96–1.13; 0.2865)	0.96 (0.87–1.07; 0.4421)
*Streptococcus* gr.A	1.61 (0.58–4.48; 0.3594)	1.73 (1.31–2.29; 0.0001) *	0.93 (0.33–2.59; 0.8909)
*Enterococcus faecalis*	0.98 (0.95–1.02; 0.3173)	0.98 (0.95–1.02; 0.3173)	1.00 (0.61–1.67; 0.9891)
*Serratia marcescens*	1.01 (0.95–1.07; 0.8052)	1.08 (0.94–1.24; 0.2655)	0.93 (0.81–1.07; 0.3311)
*Enterobacter* spp.	0.93 (0.83–1.03; 0.1577)	1.65 (1.23–2.22; 0.0009) *	0.56 (0.42–0.75; 0.0001) *
*Acinetobacter baumannii*	1.25 (0.95–1.65; 0.1089)	2.22 (1.50–3.28; 0.0001) *	0.56 (0.36–0.89; 0.0137) *
*Stenotrophomonas maltophilia*	1.50 (1.11–2.02; 0.0077) *	1.51 (1.08–2.10; 0.0148) *	0.99 (0.70–1.42; 0.9697)
*Enterococcus faecium*	1.00 (0.59–1.66; 0.9951)	1.05 (0.99–1.11; 0.0833)	0.95 (0.90–1.01; 0.0833)
*Klebsiella* spp.	1.09 (0.92–1.29; 0.3175)	1.00 (0.53–1.92; 0.9914)	1.09 (0.92–1.29; 0.3175)
Other	1.53 (1.00–2.34; 0.0492) *	1.62 (0.89–2.96; 0.1165)	0.94 (0.47–1.91; 0.8741)
*Sphingomonas paucimobilis*	1.56 (0.38–6.36; 0.5385)	NA	NA
*Enterococcus* spp.	0.92 (0.73–1.16; 0.4847)	1.00 (0.81–1.24; 0.9933)	0.92 (0.80–1.07; 0.2722)
*Chryseobacterium* spp.	1.00 (0.38–2.48; 0.9882)	NA	NA
*Morganella* spp.	1.50 (0.67–3.34; 0.3206)	1.00 (0.33–3.16; 0.9921)	1.50 (0.67–3.34; 0.3206)
*Acinetobacter* spp.	2.13 (0.83–5.50; 0.1168)	1.09 (0.71–1.69; 0.6950)	1.96 (0.76–5.03; 0.1643)
*Citrobacter* spp.	0.80 (0.42–1.54; 0.5025)	0.74 (0.43–1.28; 0.2830)	1.08 (0.62–1.88; 0.7936)
*Streptococcus* spp.	0.88 (0.34–2.22; 0.7789)	0.90 (0.50–1.63; 0.7319)	0.97 (0.41–2.33; 0.9466)
*Candida parapsilosis*	NA	NA	NA
*Salmonella* spp.	1.00 (0.19–5.37; 1.0000)	3.00 (1.47–6.14; 0.0026) *	0.33 (0.16–0.68; 0.0026) *
*Streptococcus* gr.B	1.33 (0.58–3.09; 0.5017)	0.80 (0.52–1.24; 0.3183)	1.67 (0.81–3.41; 0.1618)
*Pseudomonas* spp.	0.75 (0.32–1.74; 0.5017)	0.90 (0.40–2.00; 0.7963)	0.83 (0.47–1.47; 0.5277)
*Candida albicans*	2.11 (0.19–23.03; 0.5387)	2.21 (0.20–24.10; 0.5144)	0.96 (0.06–15.12; 0.9743)
*Serratia* spp.	1.50 (0.85–2.64; 0.1601)	NA	NA
*Candida* spp.	NA	0.38 (0.04–3.92; 0.4129)	NA
*Candida tropicalis*	NA	NA	NA
*Proteus* spp.	0.20 (0.03–1.15; 0.0720)	0.20 (0.03–1.15; 0.0720)	1.00 (0.08–8.73; 0.9805)
*Haemophilus influenzae*	NA	NA	NA

*: statistically significant, NA: not applicable.

**Table 7 antibiotics-14-00411-t007:** Rate comparison for all XDR organisms.

IRR (95%CI; *p*)	2019 vs. 2021	2019 vs. 2023	2023 vs. 2021
Overall XDR	1.47 (1.17–1.86; 0.0006) *	1.43 (1.18–1.74; 0.0002) *	1.03 (0.81–1.31; 0.8394)
*Klebsiella pneumoniae*	1.62 (1.14–2.31; 0.0074) *	0.93 (0.74–1.17; 0.5390)	1.74 (1.23–2.47; 0.0018) *
*Pseudomonas aeruginosa*	1.09 (0.70–1.68; 0.7121)	1.04 (0.68–1.59; 0.8577)	1.04 (0.66–1.64; 0.8514)
CoNS	1.11 (0.67–1.84; 0.6947)	1.50 (0.98–2.30; 0.0603)	0.74 (0.44–1.23; 0.2439)
*Escherichia coli*	1.76 (0.91–3.41; 0.0933)	1.75 (0.97–3.16; 0.0608)	1.00 (0.49–2.07; 0.9931)
*Serratia marcescens*	4.04 (1.51–10.77; 0.0054) *	2.22 (0.87–5.69; 0.0968)	1.82 (0.50–6.57; 0.3617)
*Acinetobacter baumannii*	NA	NA	NA
*Enterobacter* spp.	0.71 (0.28–1.84; 0.4836)	0.80 (0.29–2.17; 0.6577)	0.89 (0.37–2.18; 0.8064)
*Morganella* spp.	NA	NA	NA
Other	NA	NA	NA
*Proteus mirabilis*	1.53 (0.19–12.51; 0.6919)	3.82 (0.46–31.94; 0.2156)	0.40 (0.03–6.16; 0.5113)
*Chryseobacterium* spp.	1.56 (0.52–4.69; 0.4329)	NA	NA
*Klebsiella* spp.	0.43 (0.10–1.83; 0.2522)	1.93 (0.22–17.14; 0.5556)	0.22 (0.03–1.89; 0.1688)
*Pseudomonas* spp.	NA	NA	NA
*Acinetobacter* spp.	NA	NA	NA
*Staphylococcus aureus*	0.71 (0.04–11.27; 0.8070)	0.96 (0.06–15.31; 0.9776)	0.74 (0.05–11.73; 0.8289)

*: statistically significant, NA: not applicable.

**Table 8 antibiotics-14-00411-t008:** Rates of abandonment.

		2019 vs. 2021	2019 vs. 2023	2023 vs. 2021
Region	IRR	0.73	0.55	1.32
	95%	0.43–1.22	0.32–0.93	0.76–2.28
	*p*	0.1982	0.0186 *	0.2995
County	IRR	0.57	0.30	3.31
	95%	0.17–1.90	0.98–0.88	1.14–10.24
	*p*	0.3018	0.0166 *	0.0166 *
Hospital total	IRR	1.70	0.61	3.29
	95%	0.31–17.27	0.17–2.42	0.54–34.53
	*p*	0.5505	0.4011	0.1591
Hospital infected	IRR	NA	0.62	NA
	95%	NA	0.10–4.21	NA
	*p*	NA	0.5379	NA

*: statistically significant, NA: not applicable.

**Table 9 antibiotics-14-00411-t009:** Antibiotic class resistance analysis for abandoned patients.

	2019			2023		
	Abandonment	No Abandonment	IRR (95% CI; *p*)	Abandonment	No Abandonment	IRR (95% CI; *p*)
Penicillins	50.00%	41.73%	1.20 (0.64–2.23; 0.5686)	40.00%	41.17%	0.97 (0.33–2.85; 0.9581)
Cephalosporins	40.00%	27.13%	1.47 (0.69–3.16; 0.3180)	60.00%	21.22%	2.83 (1.38–5.81; 0.0046) *
Combination therapy	0.00%	24.63%	NA	20.00%	24.12%	0.83 (0.14–4.79; 0.8342)
Carbapenems	10.00%	11.16%	0.90 (0.14–5.77; 0.9080)	20.00%	6.93%	2.89 (0.50–16.75; 0.2375)
Fluoroquinolones	50.00%	26.69%	1.87 (1.00–3.49; 0.0483) *	0.00%	31.09%	NA
Aminoglycosides	60.00%	29.84%	2.01 (1.21–3.35; 0.0072) *	20.00%	20.60%	0.97 (0.17–5.61; 0.9737)
Macrolides	20.00%	12.90%	1.55 (0.45–5.38; 0.4895)	20.00%	10.89%	1.84 (0.32–10.64; 0.4975)
Lincosamides	20.00%	9.46%	2.11 (0.61–7.35; 0.2389)	0.00%	5.87%	NA
Cyclines	0.00%	8.01%	NA	20.00%	6.52%	3.07 (0.53–17.81; 0.2116)
Glycopeptides	10.00%	4.53%	2.21 (0.34–14.30; 0.4060)	0.00%	0.73%	NA
Urinary	10.00%	9.46%	1.06 (0.16–6.81; 0.9536)	60.00%	17.45%	3.44 (1.67–7.07; 0.0008) *
Reserve antibiotics	70.00%	21.07%	3.32 (2.20–5.02; 0077) *	20.00%	22.10%	0.90 (0.16–5.23; 0.9112)
Other antibiotics	0.00%	5.54%	NA	20.00%	18.04%	1.11 (0.19–6.41; 0.9083)

*: statistically significant, NA: not applicable.

**Table 10 antibiotics-14-00411-t010:** Antibiotic class resistance and phenotype logistic regression.

Variable	Coefficient	Std. Error	Wald	*p*	Odds Ratio	95% CI
Cephalosporines	1.64	0.76	4.69	0.0304 *	5.17	1.17–22.89
Fluoroqinolones	−0.92	0.76	1.48	0.2242	0.39	0.09–1.76
Amynoglicozide	1.18	0.69	2.96	0.0853	3.27	0.85–12.56
Urinary antibiotics	1.2	0.67	3.16	0.0755	3.31	0.88–12.43
Reserve antibiotics	1.73	0.62	7.9	0.0049 *	5.64	1.69–18.84
ESBL	−1.75	0.91	3.72	0.0536	0.17	0.03–1.03
CRO	0.75	0.92	0.66	0.4168	2.12	0.35–12.94
MDR	−1.57	1.03	2.35	0.1254	0.21	0.03–1.55
XDR	0.34	0.96	0.13	0.7235	1.40	0.22–9.19
Constant	−6.86	0.46	223.16	<0.0001 *		
Overall fit *p*	0.0119 *					
R^2^	0.0031					

*: statistically significant, NA: not applicable.

## Data Availability

Data available upon request from the corresponding author.

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
