# Peer review of "The Impact of the COVID-19 Pandemic on Pediatric Microbial Resistance Patterns and Abandonment Rates in Western Romania—An Interdisciplinary Study"

_antibiotics, 2025, doi:10.3390/antibiotics14040411_

Round 1

Reviewer 1 Report

Comments and Suggestions for Authors
  • Rewrite the abstract to fit the paper
  • Highlight the significance in better way in the introduction 
  • Were CLSI or EUCAST guidelines followed?

  • How were strains identified and confirmed (e.g., MALDI-TOF, PCR, etc.)?

  • Add more for sampling techniques
  • Write details for ethical statement and informed consent
  • Elaborate more in the discussion section
  • Some references are very old, more than 7 years. Need to be updated 

Author Response

Answer to Reviewer 1

Thank you very much for reviewing our article and for all your recommendations, we took all of them into consideration, and now we hope that the quality and the clarity of the present paper was improved, making it a more valuable one.

In response, here are punctually our answers to your recommendations.

  • Rewrite the abstract to fit the paper

Thank you. This was addressed.

  • Highlight the significance in better way in the introduction 

Thank you, this was addressed in a separate paragraph in the introduction.

  • Were CLSI or EUCAST guidelines followed?

Yes, this is mentioned in the methods and limitation sections. Specifically, CLSI norms were used for 2019 and 2021 and EUCAST for 2023. This shift is a normal trend in Europe for some time and a requirement for certain accreditation systems in our country. As there may be some differences between the two organizations, this was also mentioned as a potential limitation.

  • How were strains identified and confirmed (e.g., MALDI-TOF, PCR, etc.)?

Thank you for noticing this oversight, this information was added.

  • Add more for sampling techniques

Thank you. This was added.

  • Write details for ethical statement and informed consent

Ethics was rewritten.

  • Elaborate more in the discussion section

Thank you, this was also addressed alongside a similar answer for Reviewer 2.

  • Some references are very old, more than 7 years. Need to be updated 

Thank you for noticing this. With the exception of Magiorakos et al., which is still relevant today, older references are in regards to pediatric abandonment in medical units. Unfortunately, after extensive search more updated articles on this topic are scarce, especially those explicitly mentioning the role of infections in pediatric abandonment in the hospital/ maternity. To address this situation properly, a paragraph was added in the limitations section regarding the need for renewed attention on this topic.

Reviewer 2 Report

Comments and Suggestions for Authors

The authors have conducted an interesting study with the aim of analyzing the trends in pediatric antibiotic resistance patterns in Romanian hospitals before, during, and after the Sars-CoV-2 pandemic. This is an intriguing manuscript that addresses, among other objectives, the issue of antibiotic resistance, one of the most significant social problems of the last century. Below are my comments.

In the abstract, please explicitly state the main aim of the study.

Line 60 – Are there available data regarding the pediatric mortality and morbidity rates due to COVID-19 in the authors' country? If possible, it would be interesting to include this information.

Line 134 – Please specify the type of infection being referred to. What type of microbiological tests do these results pertain to? This information becomes clear further on in the manuscript (see line 234) but should be specified earlier.

The statistical analysis is very interesting and appropriate for the authors' purpose. It would be interesting to analyze the prevalence of antibiotic resistance not only among different bacteria but also between different microbiological methods performed (e.g., urine culture vs. blood culture). If feasible, this could be a good addition to the manuscript or a suggestion for future studies by the authors, depending on data availability.

The discussion is adequate. According to the authors, what are the best strategies to reduce the rate of antibiotic-resistant infections in the pediatric population?
For instance, Pseudomonas and other urinary infections in the pediatric population have been shown to respond optimally to short-course antibiotic therapy (see 10.3390/children9111647), achieving therapeutic efficacy comparable to standard regimens while reducing the incidence of antibiotic resistance and reinfection. This concept is well applicable to UTIs but is more challenging to implement in chronic patients, such as those with cystic fibrosis, or in cases of serious bacterial infections like meningitis or sepsis.

Minor improvements are needed in the English translation of the manuscript.

I thank the authors for their excellent work and look forward to reviewing the revised version of the paper.

Comments on the Quality of English Language

Minor improvements are needed in the English translation of the manuscript.

Author Response

Answer to Reviewer 2

Thank you very much for reviewing our article and for all your recommendations, we took all of them into consideration, and now we hope that the quality and the clarity of the present paper was improved, making it a more valuable one.

In response, here are punctually our answers to your recommendations.

  1. In the abstract, please explicitly state the main aim of the study.

Thank you, this was addressed.

  1. Line 60 – Are there available data regarding the pediatric mortality and morbidity rates due to COVID-19 in the authors' country? If possible, it would be interesting to include this information.

Thank you, this was addressed in a separate paragraph in the introduction, along a reference from our national institute for statistics.

  1. Line 134 – Please specify the type of infection being referred to. What type of microbiological tests do these results pertain to? This information becomes clear further on in the manuscript (see line 234) but should be specified earlier.

Thank you, this was addressed.

  1. The statistical analysis is very interesting and appropriate for the authors' purpose. It would be interesting to analyze the prevalence of antibiotic resistance not only among different bacteria but also between different microbiological methods performed (e.g., urine culture vs. blood culture). If feasible, this could be a good addition to the manuscript or a suggestion for future studies by the authors, depending on data availability.

Thank you for the idea. The authors acknowledge the need of differential diagnosis based on sample. However, at this stage, it would take very long for an already extensive article. We would like to mention that such research is planned, as part of the doctoral studies of the first author and will be published at a later time.

  1. The discussion is adequate. According to the authors, what are the best strategies to reduce the rate of antibiotic-resistant infections in the pediatric population?
    For instance, Pseudomonas and other urinary infections in the pediatric population have been shown to respond optimally to short-course antibiotic therapy (see 10.3390/children9111647), achieving therapeutic efficacy comparable to standard regimens while reducing the incidence of antibiotic resistance and reinfection. This concept is well applicable to UTIs but is more challenging to implement in chronic patients, such as those with cystic fibrosis, or in cases of serious bacterial infections like meningitis or sepsis.

Thank you for the suggestion. This was addressed, along with a similar request from Reviewer 1.

  1. Minor improvements are needed in the English translation of the manuscript.

English was revised by a C1 level speaker.

Round 2

Reviewer 1 Report

Comments and Suggestions for Authors

All required changes were carried out

Reviewer 2 Report

Comments and Suggestions for Authors

The authors have adequately addressed all my comments. The manuscript has significantly improved compared to the previous version. I have no further comments and believe it can be accepted, pending the editor's final decision.